# Lynch-like Syndrome: Potential Mechanisms and Management

**DOI:** 10.3390/cancers14051115

**Published:** 2022-02-22

**Authors:** Alejandro Martínez-Roca, Mar Giner-Calabuig, Oscar Murcia, Adela Castillejo, José Luis Soto, Anabel García-Heredia, Rodrigo Jover

**Affiliations:** 1Servicio de Medicina Digestiva, Hospital General Universitario de Alicante, Instituto de Investigación Sanitaria, ISABIAL, Universidad Miguel Hernández, 03010 Alicante, Spain; martinez_aleroc@gva.es (A.M.-R.); mar.giner-calabuig@yale.edu (M.G.-C.); omp_89@hotmail.com (O.M.); garcia_anabelher@gva.es (A.G.-H.); 2Digestive Disease Department, University of Yale, New Heaven, CT 06520, USA; 3Laboratorio de Genética Molecular, Hospital General Universitario de Elche, FISABIO, 03203 Elche, Spain; castillejo_ade@gva.es (A.C.); soto_jos@gva.es (J.L.S.)

**Keywords:** lynch syndrome, lynch-like syndrome, hereditary cancer, colorectal cancer, DNA mismatch repair genes

## Abstract

**Simple Summary:**

Lynch-like syndrome (LLS) is defined as colorectal cancer cases with microsatellite instability (MSI) and loss of expression of MLH1, MSH2, MSH6, or PMS2 by immunohistochemistry (IHC) in the absence of a germline mutation in these genes that cannot be explained by *BRAF* mutation or MLH1 hypermethylation. The application of the universal strategy for the diagnosis of Lynch syndrome (LS) in all CRCs is leading to an increase in the incidence of cases of LLS. It has been described that risk of cancer in relatives of LLS patients is in between of that found in Lynch syndrome families and sporadic cases. That makes LLS patients and their families a challenging group for which the origin of CRC is unknown, being a mixture between unidentified hereditary CRC and sporadic cases. The potential causes of LLS are discussed in this review, as well as methods for identification of truly hereditary cases.

**Abstract:**

Lynch syndrome is an autosomal dominant disorder caused by germline mutations in DNA mismatch repair (MMR) system genes, such as *MLH1*, *MSH2*, *MSH6*, or *PMS2*. It is the most common hereditary colorectal cancer syndrome. Screening is regularly performed by using microsatellite instability (MSI) or immunohistochemistry for the MMR proteins in tumor samples. However, in a proportion of cases, MSI is found or MMR immunohistochemistry is impaired in the absence of a germline mutation in MMR genes, *BRAF* mutation, or *MLH1* hypermethylation. These cases are defined as Lynch-like syndrome. Patients with Lynch-like syndrome represent a mixture of truly hereditary and sporadic cases, with a risk of colorectal cancer in first-degree relatives that is between the risk of Lynch syndrome in families and relatives of sporadic colon cancer cases. Although multiple approaches have been suggested to distinguish between hereditary and sporadic cases, a homogeneous testing protocol and consensus on the adequate classification of these patients is still lacking. For this reason, management of Lynch-like syndrome and prevention of cancer in these families is clinically challenging. This review explains the concept of Lynch-like syndrome, potential mechanisms for its development, and methods for adequately distinguishing between sporadic and hereditary cases of this entity.

## 1. Introduction

Lynch-like syndrome (LLS) is defined as colorectal cancer cases with microsatellite instability (MSI) and loss of expression of MLH1, MSH2, MSH6, or PMS2 by immunohistochemistry (IHC) in the absence of a germline mutation in these genes that cannot be explained by *BRAF* mutation or *MLH1* hypermethylation [1]. Managing these cases is challenging because the subsequent carcinogenic process is yet to be unveiled. LLS is probably caused by somatic mutations in the mismatch repair (MMR) genes, and, therefore, it is sporadic [2,3]. However, patients with LLS and their relatives have an increased risk of colorectal cancer (CRC), suggesting a possibility of inherited risk. Thus, the most probable scenario is that LLS represents a mixture of sporadic MSI cases, unidentified Lynch syndrome (LS) cases, and possibly other hereditary cases of yet-to-be-determined origin [1,4]. Differentiating between both sporadic and hereditary origin has been a challenge, mainly due to the difficulty in conducting mutational somatic studies of CRC samples. In this review, the characteristics of LLS cases, the potential causes, and recommendations on managing these cases are discussed.

## 2. Carcinogenic Pathways in Colorectal Cancer

CRC is the third most commonly diagnosed cancer after breast and lung cancer, accounting for 10% of all cancers diagnosed worldwide, or approximately 2 million cases every year [5]. Approximately 70–80% of CRC are considered to be sporadic [6].

Based on molecular characteristics, CRC can be subclassified into tumors that exhibit chromosomal instability (CIN) and tumors with MSI. CIN includes structural abnormalities or changes in the number of chromosomes [7]. Missegregation of chromosomes can lead to both the activation of oncogenes, such as *KRAS*, and tumor-suppressor effects, such as inactivation of *APC* [8,9]. Approximately 70–80% of colorectal tumors have chromosomal abnormalities associated with poor prognosis [10]. CIN causes tumor progression by increasing genomic alterations, making the tumor more aggressive and drug-resistant [11]. Most tumors with CIN are microsatellite stable (MSS) [12].

Between 20% and 30% of CRCs arise through the serrated pathway of carcinogenesis [13]. The serrated pathway includes tumors with MSS and MSI. Serrated polyps are the precursor lesion in this pathway. A majority of serrated polyps (80–90%) are benign lesions, with only a minority developing dysplasia. However, after dysplasia, carcinogenesis is accelerated and CRC develops [14]. Most CpG island methylator phenotype (CIMP) tumors are MSS or have low MSI, but if hypermethylation affects *MLH1*, CIMP tumors can also have high MSI. *BRAF* mutation is the original somatic event in this pathway, with CIMP as an early feature [13].

Finally, tumors with MSI represent 15% of all CRCs and are MMR-deficient (MMR-D) [15,16,17]. The role of MMR system is to correct mismatches and small insertion/deletions that occur during DNA replication. Microsatellites are repetitive sequences that are prone to the accumulation of errors [18]. When there is a deficiency in the MMR machinery, these errors cannot be corrected, resulting in microsatellites that are different in size and MSI [19]. On the one hand, the most frequent cause of MMR-D is the hypermethylation of *MLH1*, which occurs in 80% of tumor with MSI tumors, as a consequence of CIMP [20]. On the other hand, in a percentage of the remaining cases, MMR-D is caused by germline mutations in the MMR genes *MLH1*, *MSH2*, *MSH6*, and *PMS2* [21]. Currently, three CRC phenotypes are related to the presence of MSI: MSI sporadic tumors, LS, and LLS (Figure 1).

## 3. Lynch Syndrome

LS is the most common hereditary cancer syndrome and accounts for approximately 3% of all CRCs [22]. LS is an autosomal dominant disorder caused by germline mutations in *MLH1*, *MSH2*, *MSH6*, and *PMS2*, as well as deletions in *EPCAM*. Germline *EPCAM* deletions result in methylation of the surrounding genomic region, affecting the *MSH2* promoter located 18 Kb downstream. As a consequence, *MSH2* gene expression is silenced [23]. Constitutional epigenetic silencing of *MLH1* [24,25,26,27,28,29,30,31,32,33,34,35,36] and hypermethylation of *MSH2* as a consequence of *EPCAM* deletion [29] have been rarely reported in some families.

LS patients develop multiple tumors, most frequently colorectal and endometrial [37], but also upper gastrointestinal, ovarian, biliary, urinary, brain, non-melanoma skin, and prostate tumors [38]. LS patients are diagnosed at an early age; with a mean age of diagnosis of around 45 years, they develop cancer a mean of 23 years earlier than the general population [39]. Lynch tumors develop faster than sporadic CRC [40]. LS patients have an increased risk of synchronous and metachronous neoplasias. Approximately 7% of LS patients have multiple cancers when diagnosed [40,41]. LS tumors are poorly differentiated, and some present with mucinous features, a medullary growth pattern or showing infiltrating lymphocytes [42,43]. Moreover, their location is predominately in the proximal colon [40,44].

The diagnostic algorithm for LS starts by testing tumors for MSI and/or loss of immunochemical expression of MMR proteins. The Jerusalem guidelines, so-called ‘universal screening’, recommend screening all CRCs and endometrial patients <70 years old for MSI or MMR-D [45]. If MLH1 is lost in IHC, the tumor should then be tested for methylation of the promoter of *MLH1* and/or the *BRAF* V600E mutation to rule out sporadic CIMP tumors. If testing negative, patients are submitted to germline testing, which includes the sequencing and the analysis of deletions and duplications in the appropriate MMR genes. Germline testing results confirm an LS diagnosis [46,47], whereas, if *MLH1* is methylated in a tumor, a complementary *MLH1* methylation study in blood should be performed to identify constitutional epimutation of *MLH1* [48] (Figure 2).

## 4. Lynch-like Syndrome

Up to 50% of cases of suspected LS in patients with CRC that test positive for MMR-D by IHC or MSI do not have any germline mutation in an MMR gene, *BRAF* alteration or *MLH1* hypermethylation [1]. These cases are defined as Lynch-like syndrome (LLS). LLS possibly describes a heterogeneous group of conditions that possibly includes a mix between sporadic and hereditary cases. Although multiple approaches have been suggested to distinguish between hereditary and sporadic cases, accurate testing and a consensus on the adequate classification of these patients are still lacking.

### 4.1. Demographics

The risk of developing CRC and other LS-related cancers in LLS cohorts and their first-degree relatives is lower than in LS patients. However, LLS patients are more likely to have CRC than sporadic cases [1,44,50,51]. Moreover, the age at diagnosis follows the same pattern, and it is higher than in LS but lower than in sporadic cases [1,52]. LLS tumors share the pathological characteristics of MSI CRC, as they are mainly located on the proximal colon, frequently have a large size, and usually present a higher concentration of infiltrated lymphocytes [1,4,44]. On the other hand, no differences in sex have been found. Some series show a predominance of female patients, but without significant differences [1,53].

### 4.2. Family History

LLS patients can be clinically differentiated into two groups. One group includes patients with family history that suggests a hereditary origin. These families probably have a hereditary factor that predisposes them to a high risk of CRC. However, in these cases, the genetic alteration is unknown. Furthermore, there are other LLS families who do not have a history of cancer. In that case, it is possible to find a double somatic mutation in MMR genes that explains their MSI. These cases could be considered sporadic [4,52,54]. As noted previously, LLS cases are a heterogeneous group that includes sporadic and inherited cases, and it is necessary to define molecular tools to efficiently differentiate between both groups. Moreover, there are no differences in the clinical or pathological characteristics that differentiate between hereditary and sporadic cases [54]. The validation and implementation of molecular analysis of MMR genes in tumors as a part of routine diagnosis is still a challenge in many laboratories.

### 4.3. Pathology and IHC

All CRCs undergo IHC for MMR proteins MLH1, MSH2, MSH6, and PMS2 and/or MSI testing [45]. Tumors are considered MMR-D when they exhibit a loss of expression of the MMR proteins by IHC. Universal IHC or MSI testing increases the detection of LLS [53]. IHC has shown that the main defective protein in LLS tumors is MLH1 [55]. Picó et al., found that 50% of LLS patients lack MLH1/PMS2 and 27.9% lack MSH2/MSH6 expression [54]. These results are similar to previous studies that obtained the same data. Perez-Carbonell et al., identified a lack of MLH1 protein in 29 of 62 patients, followed by MSH2 and MSH6 loss (19 patients), and Overbeek et al., exposed MLH1 deficiency in 5 of 16 families [51,53]. Although one publication reported that grade 1 dysplasia was predominant in LLS, in contrast to grade 3 in LS [11,14], no other histological characteristics have been found [4,56].

### 4.4. Cancer Risk

First-degree relatives of patients with LLS CRC have an increased risk of CRC and non-CRC LS-related cancers. However, this risk is lower than that found in LS families. This can be seen in Rodriguez-Soler et al., who estimated a standardized incidence ratio (SIR) for CRC of 2.12 in LLS cases versus 6.04 in LS patients. In addition, Picó et al., estimated an SIR for CRC (4.25 in LS vs. 2.08 in LLS) and LS-associated neoplasms (5.01 in LS vs. 2.04 in LLS) [1,4,50]. Regarding non-CRC cancers associated with LS, they estimated an SIR of 1.69 in LLS families vs. 2.81 in LS patients [1,52]. These results are supported by Win et al., who showed that the risk of a first-degree relative of an LS patient developing CRC (hazard ratio (HR) = 5.37) is higher than in LLS (HR = 2.06) and MMR-D non-LS groups (HR = 1.04) (50). LS families also have a higher risk of developing endometrial tumors. Other studies have shown an excess risk of pancreatic cancer in LLS families [4]. Due to this increased risk of developing CRC in LLS families, if it is not possible to safely identify truly sporadic cases, LLS families should be considered a high-risk group, and some authors suggest screening colonoscopies every 3 years for first-degree relatives of LLS patients [57].

## 5. Potential Causes of Lynch-like Syndrome

Different plausible causes to explain the origin of LLS tumors have been described. According to the hereditary origin, unknown mechanisms or germline mutations in other genes than those involved in the classical MMR system could mimic the Lynch phenotype with MMR-D. In addition, some LLS cases could be LS with unidentified germline MMR mutations. In contrast, LLS could be due to somatic defects in genes related to tumor onset and progression or due to biallelic alterations in MMR genes outside *MLH1* promoter methylation [2,3,58], thus having a sporadic origin. A frequent explanation for LLS cases that should always be ruled out is false-positive IHC/MSI results, which represent approximately 19% of cases in some series [59], and confirmation of MSI and IHC status should be the first step before classifying a case as LLS. Figure 3 describes different potential causes of LLS. It is important to clarify that, if some LLS cases after their molecular analysis can be classified in another category, they will no longer be considered LLS. They will be included in the surveillance program of the new group.

### 5.1. Germline Mutations in Other Genes Affecting the MMR System

The fact that LLS patients are younger at diagnosis than sporadic cases and some of them have a family history of LS-related neoplasias suggests that germline mutations in other genes could also be involved in cancer development in some of these cases (Figure 3). It is important to distinguish whether MMR-D is driving tumor formation or is a secondary event. Germline mutations in *MUTYH* and *POLE* have been reported in some patients with MMR-D [31,60,61,62]. Mutations in *MUTYH* have been previously associated with *MUTYH*-attenuated polyposis [31,62]. In addition, mutations in *MUTYH* have been described in *MLH1*-methylated tumors [31,62]. Approximately 1–3% of LLS cases carry biallelic mutations in *MUTYH* [31,62]. In addition, mutations in the exonuclease domain of *POLE* and *POLD1* cause a hypermutator phenotype that confers a high predisposition to developing attenuated colorectal polyposis at an early age. *POLE* and *POLD1* mutations may be associated with MMR-D in some cases due to MMR mutations secondary to the hypermutator phenotype [60,61,63,64,65].

On the other hand, Xavier et al., found potentially pathogenic variants in a group of genes involved in the regulation of cellular activity (*EXO1*, *POLD1*, *RFC1*, and *RPA1*) [55]. *EXO1* is related to the union of MLH1 and MSH2, and a mutation in these genes may trigger MMR-D [66]. In addition, *RPA1* and *POLD1* are associated with harmful effects in tumors with mutations in these genes [67,68]. *RFC1* has been described in the development of different malignancies. Huang et al., noted the presence of a variation of this gene in a plasmatic cell tumor [69]. Moreover, somatic mutations in *RFC1* were reported in 10.2% of uterine carcinomas and 5.5% of CRCs [70]. This gene also plays an important role in genomic integrity because it is a member of the *BRCA1*-associated genomic surveillance complex [71]. Golubicki et al., found unknown variants in four genes (*POLE*, *ERCC6*, *RAD54L*, and *PALB2*) in a group of LLS patients [72]. *ERCC6* and *PALB2* have been associated with CRC [61,73,74], and the *PALB2* variant was previously reported in a suspected case of LS [75].

Next-generation sequencing (NGS) studies have allowed the identification of pathogenic variants that could be candidates for familial CRC with unknown genetic basis. Recently published studies have identified pathogenic variants in genes that maintain DNA integrity resulting in a variety of clinical phenotypes. Germline variants in *NTHL1* cause adenomatous polyposis and CRC [76]; *MCM9* variants are associated with hereditary mixed polyposis, CRC, and primary ovarian failure [65,77]; and variants in *FAN1* cause hereditary CRC by impairing DNA repair [78]. Following this line of inquiry, variants in *BUB1* and *BUB3* [79], *SETD2* [80], *WRN* [81], *BARD1* [81], *MCPH1* [81], and *REV3L* [81] have been found in the germline analysis of LLS cases, linking the mutation of *WRN*, *BARD1*, *MCPH1*, and *REV3L* for the first time with CRC.

### 5.2. Hereditary Cases: Unknown Mutations in MMR Genes

In some cases, LLS patients are actually LS patients whose pathogenic variants have not been identified (Figure 3). Current techniques cannot easily identify complex and cryptic mutations. Intronic regions, structural changes such as inversions, and/or copy number variation (CNV) are rarely analyzed genetic changes but may play an important role in unveiling mutations in these patients. For example, the mutation 478 bp upstream of exon 2 in *MSH2* creates a canonical splice donor site. The pseudo-exon that is created contains a stop codon that results in a truncated protein [65,82].

Structural changes have been found in some families, such as the inversion of *MSH2* exons 1–7 in 10 families in North America [83,84,85] and the inversion of *MSH2* exons 2–6 in two families in Australia [86]. Another example of structural genetic changes is the *MLH1-LRRFIP2* fusion after a paracentric inversion of chromosome 3 [87] or deletion in that same locus [65,88]. Moreover, Hellen et al., show a retrotranspositional insertion in *PMS2* mediated by LINE-1 between exon 7 and 8 [89].

Regulatory regions of MMR genes should also be taken into account. In some cases, variants in the promoter region of MMR have been associated with reduced promoter activity or transcriptional silencing of the allele [80,90]. The accumulation of mutations in the 3′UTR of genes affects mRNA stability and, therefore, protein expression. Germline 3′ UTR mutations in *MLH1* have been associated with loss of expression [91]. On the other hand, abnormal regulation of protein expression by miRNA could cause a loss of MMR gene expression. High levels of miRNA-21 downregulate *MSH2* and *MSH6* and have been found in CRC with loss of *MSH2* expression [92]. The same has been seen with *MLH1* and miRNA-155 [92]. These examples show the importance of more extensive sequencing methods to detect complex mutations in families of patients with MMR-D and without germline mutations by routine procedures.

Somatic mosaicism could also account for some LLS cases. For instance, Sourrouille et al., described a case of somatic mosaicism in *MSH2* after de novo mutation of this gene [58]. Another study described somatic mosaicism in a woman with synchronous gynecological tumors at 44 years old. The *MLH1* mutation was only present in 20% of the allele fraction in normal tissue, but her sister and father, who were also affected with LS-related tumors, carried the same mutation [93]. A recent study reported a case of de novo somatic mosaicism in which the *MLH1* mutation was detected in the tumor and at a lower level in peripheral blood but not in any other family member [94]. Mosaicism can be detected using highly sensitive NGS with high coverage, and more genetic-driven cases could be correctly identified.

Another important factor to consider is the presence of variants of uncertain significance (VUS) in approximately 30% of cases [95]. Some of them could be pathogenic but cannot be classified due to the absence of clinical, molecular, or functional evidence. Families carrying VUS are managed based on their family history of cancer until further variant classification is available [65].

### 5.3. Somatic Alteration in Other Cancer Genes or Epigenetic Structures

Some patients with sporadic cancers do not exhibit any alteration in MMR genes but lack these proteins. In these cases, other molecular mechanisms could be leading to MMR-D and a MSI phenotype. These tumors could be due to somatic alterations in cancer genes or epigenetic events outside of the MMR system [96].

Li et al., found a relationship between the epigenetic histone marker H3K36me3 and the MMR system. H3K36me3 recruits the mismatch recognition protein hMutSα (MSH2-MSH6) onto chromatin interacting with *MSH6*. This protein and the histone methyltransferase SETD2, which also acts in the trimethylation of H3K36, are required for activation of the MMR system [97]. Li et al., demonstrated that depletion of SETD2 and/or H3K36me3 in cells resulted in an MMR-D mutator phenotype, providing a molecular explanation of tumors that are positive for MSI with the MMR-D phenotype [98]. Another important molecule is proliferation cell nuclear antigen (PCNA), which plays an important role during DNA replication. Ortega et al., showed that the phosphorylation of PCNA by epidermal growth factor receptor (EGFR) alters its interaction with MMR proteins, revealing another possible mechanism of cancer development via suppression of MMR function [99].

AT-rich interaction domain 1A (ARID1A) is mutated in a large proportion of tumors [100]. These proteins interact with MSH2, recruiting it to chromatin during DNA replication. Shen et al., demonstrated that impairment of ARID1A contributes to MMR-D [101]. In addition, a somatic exonuclease domain mutation in *POLE* would be involved in phenocopying defective MMR DNA in 25% of unexplained endometrial cancers with MSI [102].

Local inflammation also promotes genetic and epigenetic alterations in CRC [103] and has been determined to be an important factor in damage to the MMR system [104]. An increase in the concentration of proinflammatory cytokine IL-6 has been demonstrated to alter MMR function. IL6 can activate STAT3, which drives MSH3 out of the nucleus and prevents it from performing its nuclear function [105]. In the same way, high levels of reactive oxygen species (ROS) can induce DNA damage, resulting in MMR-D. Chang et al., showed that non-cytotoxic H_2_O_2_ can damage MMR complexes, triggering a reduction in these proteins [106].

Somatic methylation could also explain the MMR-D present in some LLS cases (Figure 3). Many tumor suppressor genes are methylated in sporadic cancers, including *RB* [107,108], *VHL* [109], and *BRCA1* [110], as well as *MLH1* promoter hypermethylation in sporadic CRC caused by the CIMP phenotype [111]. Recently, Buckley et al., reported an association between the methylation of SHPRH and MSI burden [112]. In addition, epimutations in *MLH1* and *MSH2* have been reported in some families [24,25,26,27,28,29,30,32,33,34,35,36], but other MMR genes can also be targets of somatic methylation [65].

### 5.4. Somatic Biallelic Alteration in MMR

Recent studies have shown that somatic mutations in MMR genes are responsible for a proportion of LLS cases [2,3,58,113,114]. In one study, 17 CRC cases with MSI were screened for somatic mutations, resulting in one out of seven MLH1-D tumors with two somatic mutations in *MLH1* and three out of eight MSH2-D tumors with two somatic mutations in *MSH2* [58]. Moreover, the combination of somatic mutation and loss of heterozygosity (LOH) as a second hit was studied in 25 tumors, finding two somatic hits in 13 tumors (8/18 in *MLH1* and 5/7 in *MSH2*) [2]. Porka et al., identified two somatic events in MMR genes in 11 out of 14 tumors and a somatic mutation and LOH in 10 out of 11 (4/10 in *MLH1*, 5/10 *MSH2,* and 1/10 *PMS2*), whereas only one tumor was characterized as having two somatic mutations (*MSH6*) [56]. Lefol et al., also looked for somatic variants, LOH, and single events in 85% [97/113] of LLS tumors (85%), but double somatic hits were found in only 63% (72/113) [114]. Another study looked for CNV in addition to mutation and LOH in 40 LLS cases; 16/24 carried double somatic hits in *MLH1* and 5/12 in *MSH2* [3]. Vargas-Parra et al., obtained the same results; they found a double somatic hit in *MSH2* and *MSH6* in five tumors from four LLS patients (80). All of these results agree with those obtained by Xicola et al., who showed a somatic mutation in one MMR allele and LOH in the other allele in 4/9 LLS cases (3/4 in *MLH1* and 1/4 in *MSH2)* [81]. It seems that the most common double somatic hit is a somatic mutation combined with LOH, followed by two somatic mutations. Based on these studies, the number of cases explained by somatic inactivation could be approximately 50% (32–82%) of LLS tumors [2,3,58]. Nevertheless, the percentage of tumors with a double somatic hit could be underestimated because some tumors also exhibit VUS or uninformative LOH, and the causative effect on the MSI phenotype is uncertain. That point is reinforced with the study of Elze et al., where they show a proportion of somatic deficient MMR tumors with somatic exon deletion that is not detectable by sequencing [115].

When comparing tumors with double somatic alterations to LS tumors, no significant histopathological difference was found [116]. Tumor sequencing is the adequate way to evaluate double somatic mutations. Therefore, tumor sequencing should be considered to clarify sporadic versus hereditary causes of unexplained MMR-D [65,117].

Different studies investigated promoter methylation of MMR genes in LLS patients. Methylation of *MSH2* was only found in one [117,118] out of 53 LLS patients with loss of expression of *MSH2* by IHC studied [80,117,118]. The *MSH6* promoter was unmethylated in 108 patients with LLS and MMR-D [80,119,120], and the same happened with the *PMS2* promoter in 100 cases with loss of expression of PMS2 or MLH1 who were negative for *PMS2* promoter methylation [121]. In summary, based on these studies, somatic promoter hypermethylation of MMR genes does not seem to be the underlying cause of MMR-D in these unexplained tumors.

Therefore, there is a subgroup of LLS that can be explained by double somatic inactivation, and these cases should probably be excluded from the LLS classification due to the probable sporadic origin. However, this approach still has some open questions, because there is no standardized universally accepted technique or protocol for differentiating these cases. Moreover, the biallelic somatic inactivation of MMR genes can also be due to any of the previously described mechanisms, some of them generated by germline genetic alterations. Classifying patients as sporadic or potentially hereditary cases should also be the subject of clinical validation by adequately comparing pedigrees, with long-term follow-up of these families in order to find differences in the incidence of CRC and other LS-related disorders. When some groups advocate for generalization of a somatic study of LLS cases, it is necessary to reach a consensus on how to perform such a study and which cases could be confidently considered as sporadic with no indication for follow-up of patients and relatives. This algorithm has not been clinically validated. Table 1 show a summary of potential causes of LLS.

## 6. Future Research

Although approximately 50–60% of LLS cases can be explained by double somatic inactivation of MMR genes [2,3,114], this does not exclude the possibility that patients carry germline variants of susceptibility that increase the risk of developing cancer. Although somatic mutational testing of tumor samples could be implemented as usual practice in select centers, this practice is still used very little in regular clinical practice. Therefore, it is important to establish firstly accurate protocols and guidelines for the study of somatic mutations in tumors and then determine whether any germline variants could trigger carcinogenesis. These discoveries could be incorporated into clinical multigene panels to improve the diagnostic algorithm for these cases.

Implementing the use of NGS technologies in the diagnostic routine will allow for the characterization of the molecular profile of patients with LLS [122]. However, a collateral effect could be finding a number of VUS with uncertain pathogenicity. This will be important to increase efforts for adequate classification of the increased number of VUS, allowing personalized diagnosis, treatment, and surveillance for LLS families, even when new alterations previously unrelated to CCR are detected. All of these advances will cost-effectively decrease mortality. Exome-based studies looking for new genes involved in LLS are ongoing in several research groups [72,123]. Approaches based on the application of simultaneous somatic and germline studies in these patients can also help differentiate between hereditary and sporadic cases. An analysis of up-front somatic testing and its applicability in clinical practice and ability to differentiate germline and somatic mutations is another interesting line for future research [124].

## 7. Conclusions

Although some recent studies suggest that LLS cases can be explained by somatic mutations [2,3,114], the fact that relatives of LLS patients have an increased risk of CRC implies that there may be an inherited risk factor [4]. Therefore, LLS is probably heterogeneous and composed of both hereditary and sporadic cases. Differentiating these cases is a current challenge in the management of high-risk conditions. The first step in the study of potential causes of LLS must be the confirmation of the diagnosis, including MSI and lack of expression of MMR proteins by IHC, without *BRAF* mutation or *MLH1* hypermethylation. There are then four potential explanations for the observed MSI and MMR-D phenotype in LLS: germline alterations in other genes; atypical germline alterations in MMR genes; somatic alterations in other genes that trigger a cascade, affecting MMR expression; and somatic biallelic inactivation of MMR genes. With recent advances in and availability of NGS, it is possible to improve somatic and germline testing and possibly distinguish which force is driving carcinogenesis in these cases. Clinical validation of these findings is also needed to improve the algorithms for adequate and safe management and genetic counselling of patients.

## Figures and Tables

**Figure 1 cancers-14-01115-f001:**
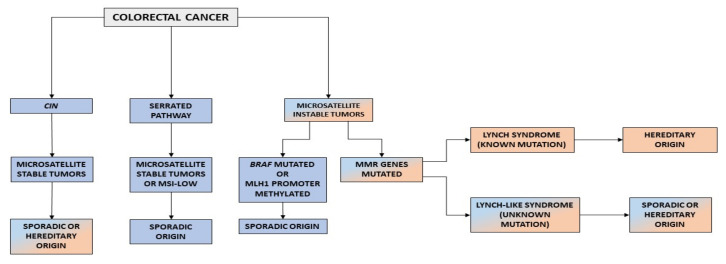
Carcinogenic pathways in CRC. CIN, chromosomal instability.

**Figure 2 cancers-14-01115-f002:**
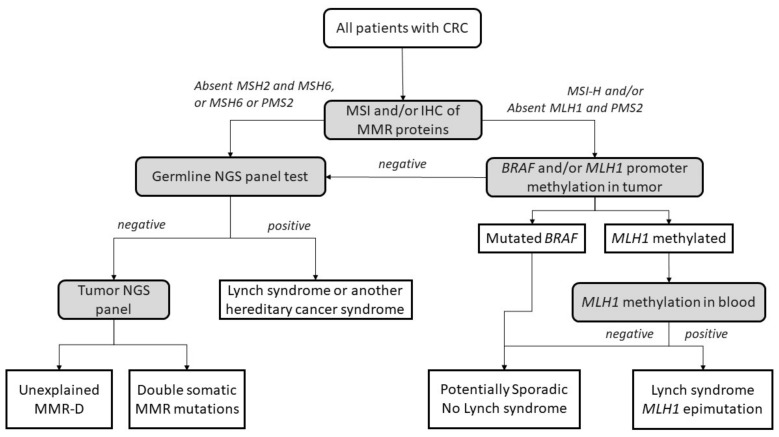
Universal screening strategy for Lynch-syndrome patients. CRC, colorectal cancer; MSI, microsatellite instability; IHC, immunohistochemistry; MMR, mismatch repair; MMR-D, mismatch repair deficiency; NGS, next-generation sequencing. Adapted from Valle et al. [49].

**Figure 3 cancers-14-01115-f003:**
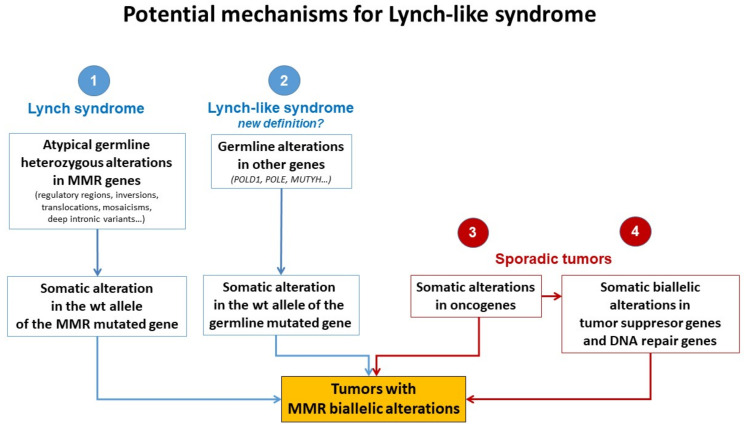
Potential mechanisms for Lynch-like syndrome. MMR, mismatch repair; wt, wild type. Adapted from Pico et al. [54].

**Table 1 cancers-14-01115-t001:** Potential causes of LLS. LLS, Lynch-like syndrome; MMR, mismatch repair.

Mutations in other Genes Affecting MMR System(Germline)	Unknown Mutations in MMR Genes(Germline)	Somatic Mutations in Cancer Genes(Somatic)	Biallelic Alteration in MMR(Somatic)
MUTYH	Mutation EXON 2 MSH2	H3K36me3	Double somatic hit
POLE/POLD1	Inversion EXON 1-7 MSH2	SETD2	Somatic mutations in MMR genes
EXO1/RFC1/RPA1	Inversion EXON 2-6 MSH2	PCNA	Methylation in MMR genes
ERCC6/RAD54L/PALB2	*MLH1*-LRRFIP2 fusion	ARID1A	
PIK3CA	*MLH1* 3′ UTR mutation	POLE	
FAN1/MCM9	Deep intronic variant in MSH2	IL-6 and oxidative stress	
NTHL1	miRNA 21 AND miRNA 155	Methylation in other genes	
BUB1/BUB3/WRN/MCPH1/REV3L	Mosaicism		
	VUS

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
