# Peer review of "Lynch-like Syndrome: Potential Mechanisms and Management"

_cancers, 2022, doi:10.3390/cancers14051115_

Round 1

Reviewer 1 Report

Review: MS for Cancers, Feb 2, 2022

Lynch-like syndrome: Potential mechanisms and management

This review is an interesting and timely manuscript on a relatively novel topic ¨Lynch-like syndrome¨, short LLS. The manuscript covers many important findings over the last years in defining patients with MSI tumors – but not having Lynch´s syndrome. As a whole well written and I only have minor comments and start from the beginning:

Definition. Perhaps the most important but still a bit unclear is the definition of LLS and it is not entirely clear from this paper what it should be – could be improved by going through it again. LLS was first defined by the same authors as patients with MSI tumors but without Lynch´s syndrome (LS) in ref.1. In simple summary it is defined as MSI tumors (MSI or IH with loss of MMR protein expression) and without germline mutations (LS) and absence of BRAF mutations (in tumor) or MLH1 hypermethylation. I think in several of the references SLL is not exactly the latter but most often simply a person with MMR deficient tumor and lack of germline mutations as was first suggested. It is not used in the same way throughout the paper and starting in Abstract where LLS is defines as in ref.1 again. Using this definition LLS can be sporadic and familial cases, have germline or somatic mutations explaining the tumor phenotype.

I think the ms would improve but clearly stating what was first the definition, and if they with this paper now have changed the definition and to what and why? This is extremely important, in particular, when starting to think about management of these patients, since it is clearly important to know if SLL is associated with increased risk or not before suggesting surveillance. The abstract ends by including both sporadic and hereditary forms in LLS. Again a definition first in Introduction – next sentence after including both sporadic and familial in LLS, states CRC cases with MMR deficient tumors wo germline mutations in MMR genes – and not explained by BRAF mutations or MLH1 hypermethylation, the new definition. Obviously MLH1 hypermethylation in tumors is considered sporadic tumors – however, also familial CRC can have hypermethylation, and even LS tumors can have (rarely) hypermethylation in tumors. Why BRAF mutations exclude LLS is not clear to me? What about KRAS or other somatic mutations?

My personal recommendation would be to keep the definition of LLS to what is said in ref1 and not change it – as a starting point for even further studies to find out the origin, sporadic (based on various criteria) or inherited and related to increased risk of relatives and need of surveillance depending on the actual risk estimated. Further subclassification of LLS for future studies based on somatic or germline mutations in genes or other somatic events or known associated data.

Figure 1. This figure is not perfect. CRC is divided in three categories, where two are based on tumor phenotype is MSS (CIN) or MSI, and the third being serrated pathway which can include both MSS and MSI (L and H) tumors – so how can this third groups of CRC patients be to belong to the serrated pathway? And the serrated pathway cannot be stated as sporadic – it is often familial.

Hypermethylation is defined as sporadic which is most often true, but can BRAF mutation be considered sporadic? I don’t think so, needs some evidence supporting this statement – or change figure. So, goes back to why I think both these criteria should not be used to modify definition.

My humble suggestion would be to explain how to select the branch serrated tumors and to remove the branch of BRAF (which is common in the serrated pathway) and hypermethylation from this figure.

Lynchs syndrome. A minor comment/question – row105 – a sign I don’t know what it means after refs (40,44)?

Figure 2. I think is adapted but again the BRAF mutation is suggested to be a sporadic case, which is not correct, it is only not likely to be LS. Perhaps modify / improve this figure and make clear that the universal screening strategy is used for finding Lynch´s syndrome, and perhaps make a more general one where the attempt is to rule out all known CRC genes which most labs do today when they use gene panels.

Cancer risk. This is most interesting the report of increased risk in general in LLS patients since it is highly relevant for management of the patients and it would be good to include any more information on what LLS patients had increased risk, if known, since they include both sporadic and familial. Was it perhaps those with a family history? In general, I lack the discussion on family history in this ms as well in many of the references actually – I find this information highly relevant.

Figure 3. I think this figure is a good attempt to clear out possible subgroups of LLS. However, I think once the explanation for the phenotype is clear, it is no longer LLS but instead, something else and should no longer be considered LLS and those patients should have the recommendations based on the type of diagnosis they have, such as LS (atypical mutations), other germline mutations (POLD1 etc), clearly somatic caused by somatic events. As long as they have no clear origine – they still belong to LLS and could be considered one kind of perhaps the same category but in time, with new knowledge, they will go into another category and have their own program to follow.

Minor: reference 118 should be corrected

In summary, I have only presented my comments and views for the authors to consider and if any thing can be agreed on there is a need to modify pieces of the ms accordingly.

Reviewer 2 Report

The authors have made an extensive review of the current knowledge of undiagnosed mismatch-repair deficient tumors. Although the paper is accurate, some items may have been missed. 

First, a generally missed germline alteration is an inversion in PMS2, it might be good to add that to paragraph 5.2 as it is relatively common (PMID: 22461402). Another paper that is good to discuss is the paper by Elze et al (PMID: 3325368) for paragraph 5.4). This paper shows that even up to 90% of unexplained MMR deficient tumors may be explained by somatic events. 

This also raises the point whether starting with germline analysis is required in especially older patients. Given the fact that most MLH1-deficient tumors are caused by somatic pathogenic variants rather than germline (even after hypermethylation testing), it may be considered to test the tumor first (after counselling) rather than start with the germline. The view of the authors on this would be appreciated.
